# Universal embeddings via Reconstruction-Augmented Contrastive Learning

## Abstract

Universal multimodal embedding models play a crucial role in tasks such as multimodal search, recommendation, and retrieval-augmented generation. However, standard contrastive learning frameworks typically optimize embeddings by pulling positives closer and pushing negatives apart, without explicitly enforcing the embeddings to preserve the rich semantics of the inputs. This often yields representations with moderate discriminability but limited semantic information, which results in suboptimal retrieval performance. In this work, we propose ReCo, a novel unified embedding framework that leverages **Re**construction-augmented **Co**ntrastive learning to learn discriminative representations enriched with semantic information. By forcing the model to reconstruct the instance semantic content solely from the encoded embeddings, ReCo produces representations that are both semantically richer and more discriminative than those learned with contrastive learning alone, yielding a unified representation space that is well-suited for embedding tasks. Extensive experiments on the MMEB benchmark and multiple cross-modal retrieval tasks demonstrate that ReCo achieves superior performance and outperforms existing state-of-the-art models. The code and model weights are available at https://anonymous.4open.science/r/ReCo-0A96.

## 1 Introduction

Multimodal embedding models aim to encode inputs from multiple modalities into unified vector representations and have been widely applied in tasks such as multimodal search (Jiang et al., 2024a), recommendation (Zhang et al., 2024b; 2025), fact verification (Wachsmuth et al., 2018), and retrieval-augmented generation (RAG) (Lewis et al., 2020; Yu et al., 2024; Tanaka et al., 2025). While advanced pre-trained vision-language models such as CLIP (Radford et al., 2021), ALIGN (Jia et al., 2021), and SigLIP (Zhai et al., 2023) achieve effective cross-modal alignment between textual and visual modalities, their inherent dual-encoder architectures—processing each modality independently—limit their ability to generate unified embeddings for complex, universal multimodal contexts (Jiang et al., 2024c; Liu et al., 2025b; Wei et al., 2024).

In recent years, the rapid development and remarkable performance of multimodal large language models (MLLMs) have driven increasing interest in MLLM-based multimodal embedding models (Zhang et al., 2024d; Liu et al., 2025b; Jiang et al., 2024c). Compared to conventional dual-encoder architectures, MLLMs naturally support interleaved text and image inputs (e.g., image-text interleaved documents), enabling the modeling of complex semantic relationships between linguistic and visual modalities (Gu et al., 2025). Furthermore, by leveraging task-specific instructions, they exhibit strong adaptability across diverse retrieval tasks, making them both flexible and efficient for multimodal embedding learning (Lin et al., 2024; Kong et al., 2025). As a representative effort, (Jiang et al., 2024c) introduces VLM2Vec together with a massive multimodal embedding benchmark (MMEB), which comprises 36 datasets across four meta-tasks with task-specific instructions; VLM2Vec acquires embeddings from the final token representation of the last MLLM layer and employs contrastive learning to adapt MLLMs to embedding models, achieving better performance than existing dual-encoder models while generalizing effectively across diverse embedding tasks.

To further explore the potential of adapting MLLMs to embedding models, we conduct a series of empirical studies and observe that the semantic richness of embeddings is positively correlated with their discriminability. We reproduce the training process of VLM2Vec (Jiang et al., 2024c) and analyze the checkpoints at different training steps along with those of our proposed ReCo.

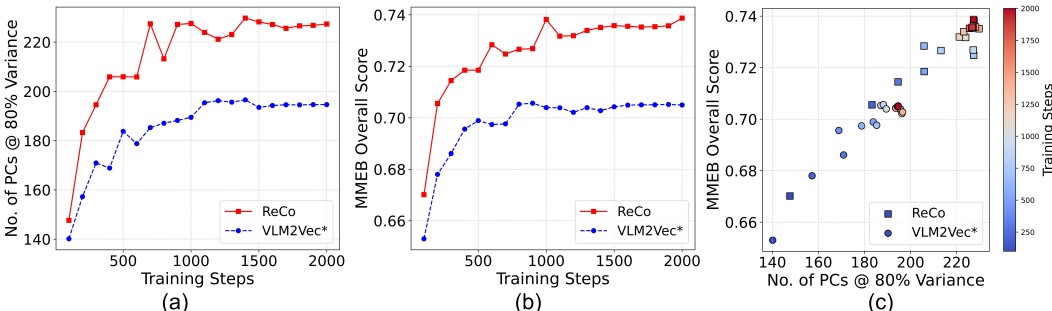

Figure 1: Positive correlation between semantic richness and discriminability of embeddings: (a) training steps vs. number (No.) of principal components (PCs) @ 80% variance, (b) training steps vs. MMEB overall score, and (c) No. of PCs @ 80% variance vs. MMEB overall score. We select the number of principal components based on the cumulative explained variance, which explains 80% of the total variance. VLM2Vec* denotes the model trained using the human–assistant conversation format.

As shown in Figure 1 (a), we observe that, as training progresses, more principal components are required to explain the same proportion (80%) of the total variance of the dataset embedding matrix, which is constructed by concatenating all sample embeddings. This suggests that variance is more evenly distributed across principal components, implying a higher effective rank of the embedding matrix (Seipel & Kalivas, 2004). In general, a higher effective rank of an embedding matrix indicates greater information capacity of the embedding space (Cover, 1999; Roy & Vetterli, 2007), enabling embeddings to capture richer input semantics (i.e., higher semantic richness), which is desirable for embedding tasks. Since the score on the MMEB dataset, which reflect the discriminability of embeddings, also increases over the course of training (Figure 1 (b)), we plot the relationship between the MMEB overall score and the number of principal components required to explain 80% of total variance as a scatter plot (Figure 1 (c)), and observe a clear positive correlation. Therefore, to improve the discriminability of embeddings, we need to further increase the semantic richness of the encoded embeddings. However, standard contrastive learning optimizes representations solely by pulling positive pairs together and pushing negative pairs apart, without explicit requirement for preserving the semantic content of the inputs. Consequently, embeddings learned from contrastive learning alone often capture limited information, leading to suboptimal retrieval performance.

Building on these observations, we propose ReCo, a novel unified embedding framework that leverages an autoregressive reconstruction task to augment contrastive learning, aiming to learn discriminative representations enriched with semantic information. Specifically, we employ contrastive learning to adapt MLLMs to embedding models, and impose an autoregressive language modeling loss to reconstruct the instance semantic content (i.e., class name for classification; answer for visual question answering; and text captions for image-to-text and text-to-image retrieval) from the embeddings. This encourages the embeddings to capture richer semantic information than contrastive learning alone, resulting in a unified representation space that is simultaneously discriminative and information-rich. In addition, to facilitate contrastive learning, we introduce a hard negative augmentation training strategy. For each batch, multiple hard negatives are sampled per instance, and a similarity-based filtering mechanism is applied during training to mitigate false-negative contamination, thereby improving the model's ability to distinguish between positive and hard negative samples. We conduct comprehensive evaluations on the MMEB benchmark and multiple cross-modal retrieval tasks. Experimental results show that ReCo consistently delivers performance improvements across all tasks, setting a new state-of-the-art over existing models. In summary, our contributions are as follows:

1. Our analysis of MLLM-based embedding model training reveals a positive correlation between the semantic richness and discriminability of embeddings, while standard contrastive learning neglects semantic preservation and thus yields embeddings with limited information and suboptimal representations.

2. To enrich the semantic information of encoded embeddings, we propose ReCo, a novel unified embedding framework that augments contrastive learning with an autoregressive reconstruction task, allowing MLLMs to produce embeddings that are both discriminative and semantically rich.

3. We introduce a hard negative augmentation training strategy that enhances contrastive learning by mining hard negatives and filtering false negatives.

4. We conduct extensive experiments on the MMEB benchmark and multiple cross-modal retrieval tasks. The results demonstrate that ReCo consistently improves performance across all evaluations and sets a new state-of-the-art in multimodal embedding.

## 2 RELATED WORK

### 2.1 MULTIMODAL LARGE LANGUAGE MODELS

MLLMs extend large language models (LLMs) by enabling them to process and integrate cross-modal information. Pioneering works such as LLaVA (Liu et al., 2023; 2024a;b), InternVL (Chen et al., 2024c;b;a; Zhu et al., 2025), MiniGPT-4 (Zhu et al., 2023), Qwen-VL (Bai et al., 2023; Wang et al., 2024; Bai et al., 2025) have demonstrated promising progress in multimodal understanding and reasoning by employing lightweight intermediate architectures to bridge visual encoders and LLM. Despite these advances, the autoregressive next-token prediction objective of MLLM inherently constrains its capacity to produce efficient multimodal representations.

### 2.2 MULTIMODAL EMBEDDING MODELS

**CLIP-series embedding models:** CLIP (Radford et al., 2021) has pioneered the foundation of multimodal embeddings, employing a dual-encoder architecture to process each modality independently and aligning image and text representations through contrastive learning. Building upon this, SigLIP (Zhai et al., 2023) enhances embedding quality and training efficiency by employing a sigmoid-based InfoNCE loss. EVA-CLIP (Sun et al., 2023) further shows the scalability of this paradigm by utilizing larger datasets and greater model capacities, thereby achieving more robust cross-modal representations. MagicLens (Zhang et al., 2024c) leverages naturally occurring image pairs and generates textual descriptions of their differences, using these as contrastive instructions to train multimodal embeddings. However, these CLIP-series models process text and image modalities independently, limiting their capacity to generate unified embeddings for complex multimodal scenarios. In contrast, recent studies have shown that multimodal large language models with early fusion of image and text features achieve better performance on multimodal embedding tasks.

**MLLM-based embedding models:** MLLM-based embedding models typically extract the hidden state of the last token in the final layer as the embeddings and employ contrastive learning to adapt MLLMs to embedding models, thereby producing highly discriminative multimodal embeddings. Building upon this general framework, recent studies have explored complementary strategies to further enhance representation quality. For instance, E5-V (Jiang et al., 2024b) shows that even unimodal training on text-only data can effectively adapt MLLMs to universal multimodal embedding models, while GME (Zhang et al., 2024d) introduces a synthetic data pipeline to leverage diverse multimodal signals and further unlock the potential of MLLMs. VLM2Vec (Jiang et al., 2024c) extends this line of work by establishing the massive multimodal embedding benchmark (MMEB), together with strong baselines trained via task instructions under a contrastive framework, providing a standard platform for systematic evaluation. Beyond advances in data and benchmarks, other methods focus on refining the learning objective itself. LLaVE (Lan et al., 2025) and QQMM-embed (Xue et al., 2025) strengthen InfoNCE training by emphasizing hard negatives—through weighting or gradient amplification—to improve robustness against challenging samples. UniME (Gu et al., 2025) leverages a powerful LLM-based teacher model to enhance the embedding capability of the MLLM's language component, while UNITE (Kong et al., 2025) seeks to harmonize the embedding space by alleviating modality competition. Complementing these approaches, B3 (Thirukovalluru et al., 2025) applies an offline clustering strategy to construct batches where samples serve as hard negatives for each other. Collectively, these efforts highlight the evolving landscape of adapting MLLMs to powerful embedding models through innovations in data design, benchmark development, and training objectives.

However, prior methods lack explicit supervision on the semantic richness of embeddings, which often results in low-information embeddings and consequently suboptimal retrieval performance. To address this limitation, we propose an autoregressive reconstruction task that reconstructs the instance semantic content from the embeddings. Together with contrastive learning, this jointly facilitates the construction of a discriminative and information-rich unified representation space.

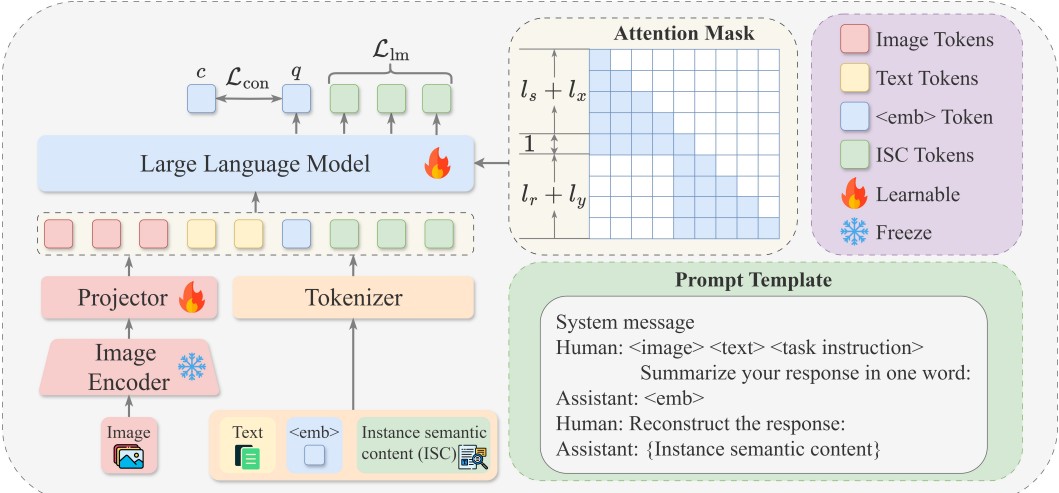

Figure 2: Overall architecture of ReCo. We propose an autoregressive reconstruction task that forces the model to reconstruct the instance semantic content from embeddings, thereby encouraging the embeddings to capture rich semantic information about the input. By jointly optimizing this task with contrastive learning, we enable MLLMs to construct a unified representation space that is both discriminative and information-rich.

## 3  ReCo

### 3.1  Multimodal Embeddings Extraction

A typical MLLM is composed of an LLM, a modality encoder, and a projector to connect them. This unified architecture enables MLLMs to flexibly process text, images, and their interleaved inputs, opening up new possibilities for unified multimodal embeddings. Prior works (Jiang et al., 2024b;c) have shown that prompt-based methods can effectively represent multimodal embeddings with MLLMs. Building on this observation, we specifically design system instructions and embedding instructions to encode multimodal inputs, as illustrated in Figure 2.

Following the conventional human–assistant conversation format of MLLMs (Liu et al., 2023), we configure a system message and feed both image and text data as human inputs. Specifically, $\langle image \rangle$ represents the image token that encodes visual information, $\langle text \rangle$ denotes textual data that needs to be encoded, and $\langle task\ instruction \rangle$ indicates task-specific instructions (see Appendix A.3 for details of different tasks). Either $\langle image \rangle$ or $\langle text \rangle$ can be omitted, allowing flexible encoding of unimodal (image or text) or multimodal (interleaved image–text) inputs. The assistant's response serves as the output, from which we extract the multimodal embeddings by taking the last layer hidden states of the $\langle emb \rangle$ token. Inspired by PromptEoL (Jiang et al., 2023) and E5-V (Jiang et al., 2024b), we adopt the Explicit One-word Limitation (EoL) strategy to enhance representation capability. Concretely, we append `"\nSummarize your response in one word:"` to the end of the human inputs, prompting the MLLM to generate embeddings of the multimodal inputs.

### 3.2  Reconstruction-Augmented Contrastive Learning

Our empirical results demonstrate a positive correlation between the semantic richness and discriminability of embeddings (see Figure 1). Based on this observation, we propose an autoregressive reconstruction task that provides explicit supervision for preserving the semantic richness of embeddings. By jointly optimizing this task with contrastive learning, we enable MLLMs to construct a discriminative and information-rich unified representation space.

#### 3.2.1  Reconstruction Targets: Instance Semantic Content

To fine-tune the MLLM for multimodal embedding tasks, we leverage the MMEB dataset (Jiang et al., 2024c), which provides task-specific instructions and spans 36 datasets across four meta-tasks—classification, visual question answering (VQA), retrieval, and visual grounding (see Appendix A.1 for details). Since the goal of representation learning is to minimize the distance be-

tween matched query–candidate pairs while maximizing the distance between mismatched pairs, we use the same instance semantic content $y$ as the reconstruction target for a user query $q$ and its corresponding positive candidate $c^+$ to align their embeddings. Specifically, we use the class name, answer, and text caption as the instance semantic content for classification, VQA, and image–text/text–image retrieval, respectively. For other tasks, no reconstruction supervision is applied.

### 3.2.2 AUTOREGRESSIVE RECONSTRUCTION TASK

For a given user input, such as a query $q$ or a candidate $c$ (denoted uniformly as $x$), the MLLM encodes it into the $\langle emb \rangle$ token to support downstream retrieval tasks. To enhance the semantic richness of the $\langle emb \rangle$ token, we design a reconstruction instruction $r$: `"Reconstruct the response:"`, which prompts the MLLM to reconstruct the corresponding instance semantic content. The input sequence to the MLLM is formed by concatenating the system instruction $s$, user input $x$, the $\langle emb \rangle$ token, the reconstruction instruction $r$, and the instance semantic content $y$, yielding a sequence of length $l = l_s + l_x + l_r + l_y + 1$, where $l_s, l_x, l_r, l_y$ denote the sequence lengths of the system instruction, user input, reconstruction instruction, and instance semantic content, respectively, and the additional 1 accounts for the $\langle emb \rangle$ token.

In the training phase, we prevent the reconstruction instruction $r$ and instance semantic content $y$ from interacting with the user input $x$, and instead require them to capture distilled multimodal information encoded in the $\langle emb \rangle$ token. To achieve this goal, a straightforward way is to modify the attention mask in each Transformer decoder layer. Based on the standard causal attention mechanism, the binary attention mask $M \in \mathbb{R}^{l \times l}$ is modified as illustrated in Figure 2. With this modified attention mask, the $y$ tokens can only attend to the $\langle emb \rangle$ token when predicting the next token, ensuring that they capture information relevant to the multimodal input exclusively through $\langle emb \rangle$. This encourages the $\langle emb \rangle$ token to retain rich semantic information about the input, yielding embeddings with rich semantic information. Following the standard formulation in (Radford et al., 2018), we supervise this task with an autoregressive language modeling objective, defined as:

$$\mathcal{L}_{\text{lm}} = -\sum_{t=1}^{l_y} \log p\big(y_t \mid \langle emb \rangle, r, y_{<t}\big) \tag{1}$$

where $y_t$ denotes the target token at position $t$ of the instance semantic content, and $p(y_t \mid \langle emb \rangle, r, y_{<t})$ represents the predicted probability distribution over the vocabulary. In this way, embeddings are explicitly guided to capture richer semantic information.

### 3.2.3 CONTRASTIVE LEARNING TASK

We adopt the InfoNCE loss for contrastive learning. Specifically, given a user query $q$ and its corresponding positive candidate $c^+$, we minimize the following contrastive loss:

$$\mathcal{L}_{\text{con}} = -\frac{1}{B} \sum_{i=1}^{B} \log \frac{\exp\big(\theta(q_i) \cdot \theta(c_i^+)/\tau\big)}{\sum_{c \in \mathcal{C}_i} \exp\big(\theta(q_i) \cdot \theta(c)/\tau\big)} \tag{2}$$

where $B$ is the batch size, $\theta(\cdot) \in \mathbb{R}^d$ denotes the normalized embedding function, and $\tau$ is the temperature hyperparameter. Ideally, the candidate set $\mathcal{C}_i$ should include all possible candidates; however, this is computationally infeasible. Therefore, mining informative negatives is crucial, as they provide richer supervisory signals for contrastive learning. In this work, we introduce a hard negative augmentation training strategy in Section 3.3 to facilitate contrastive learning.

### 3.3 HARD NEGATIVE AUGMENTATION TRAINING

Hard negatives, whose labels differ from the positives but whose embeddings are close in the representation space, are expected to provide the most utility in contrastive learning by contributing rich gradient signals. In contrast, easy negatives yield negligible gradients and thus contribute minimally to the learning process. To address this issue, we introduce a hard negative augmentation training strategy to further enhance the model's ability to discriminate between positives and hard negatives.

### 3.3.1 HARD NEGATIVE MINING

We begin by fine-tuning the MLLM-based embedding model using the contrastive loss with only in-batch samples as random negatives; i.e., $\mathcal{C}_i = (c_1^+, \ldots, c_B^+)$ in Eq. (2). For each candidate in the

batch, all candidates except $c_i^+$ are regarded as random negatives for query $q_i$. Based on the fine-tuned model, we further mine $k$ hard negatives $\hat{\mathcal{C}}_i^-$ for each query $q_i$ in the training set as follows:

$$\hat{\mathcal{C}}_i^- = \text{Select}_{K,k} \left( \theta(q_i) \cdot \theta(c) \right), \quad \text{where } c \notin \{c_i^+\} \cup \hat{\mathcal{C}}_i^h \tag{3}$$

where $c_i^+$ and $\hat{\mathcal{C}}_i^h$ denote the positive sample and the candidates whose cosine similarity with $q_i$ exceeds that of the positive sample, respectively (for classification tasks, $\hat{\mathcal{C}}_i^h = \varnothing$). Here, $\theta(q_i)$ denotes the query embeddings, $\theta(c)$ denotes candidate embeddings, and $\text{Select}_{K,k}$ represents the operation of randomly sampling $k$ candidates from the hard negative queue, i.e., the top-$K$ candidates with the highest similarity scores to $\theta(q_i)$.

### 3.3.2 FALSE NEGATIVE FILTERING

The presence of false negatives in training batches hampers the effective discrimination of hard negatives in contrastive learning. To mitigate this issue, we introduce a filtering mechanism based on a similarity threshold between candidates and their corresponding positives. During training, we consistently retain the mined hard negatives $\hat{\mathcal{C}}_i^-$ for each query $q_i$, while excluding all negatives $\hat{\mathcal{C}}_i^f$ whose similarity with $c_i^+$ exceeds the threshold $\alpha$; i.e., $\mathcal{C}_i = (\mathcal{C}_{:}^+ \cup \hat{\mathcal{C}}_{:}^-) \setminus \hat{C}_i^f$ in Eq. (2), where $\mathcal{C}_{:}^+$ and $\hat{\mathcal{C}}_{:}^-$ denote the set of all positives and all mined hard negatives within the batch, respectively.

### 3.3.3 TRAINING OBJECTIVE

The overall training objective is formulated as a combination of the autoregressive reconstruction loss and the contrastive learning loss:

$$\mathcal{L} = \lambda \cdot \mathcal{L}_{\text{lm}} + \mathcal{L}_{\text{con}} \tag{4}$$

where $\lambda$ is a weighting coefficient for the autoregressive reconstruction loss, set to 0.2 by default. At a given training iteration, the applied loss depends on the specific task. For classification, VQA, and image-text/text-image retrieval tasks, we employ the weighted sum of both losses. In contrast, for interleaved image-text retrieval and visual grounding tasks, we only apply $\mathcal{L}_{\text{con}}$.

## 4 EXPERIMENTS

### 4.1 EXPERIMENTAL SETUP

**Datasets and metrics.** In this study, we follow VLM2Vec and train our model on 20 in-distribution datasets from the massive multimodal embedding benchmark (MMEB) (Jiang et al., 2024c), which spans four meta-tasks: classification, VQA, retrieval, and visual grounding. The model is then evaluated on 20 in-distribution and 16 out-of-distribution evaluation sets from MMEB. To further validate the embedding capability of ReCo, we examine its zero-shot performance on several cross-modal retrieval tasks, including fine-grained cross-modal retrieval (Urban1K (Zhang et al., 2024a) and DOCCI (Onoe et al., 2024)) and coarse-grained cross-modal retrieval (Flickr30K (Plummer et al., 2015)). Unless otherwise specified, all results are reported in terms of Precision@1, which measures the proportion of queries for which the correct match appears at the top rank among candidates.

**Implementation details.** We adopt Qwen2-VL (Wang et al., 2024) as the backbone MLLM and train it using LoRA with rank 8 for 2000 steps, with a peak learning rate of $2 \times 10^{-5}$ and a warm-up ratio of 5%. By default, we use a total batch size of 512 and a maximum sequence length of 4096 tokens. The coefficient $\lambda$ in Eq. (4) and the temperature $\tau$ are set to 0.2 and 0.02, respectively. For classification tasks, the hard negative queue length is set to $K = 20$; for VQA, retrieval, and visual grounding tasks, we set $K = 50$. During training, we randomly sample $k = 3$ hard negatives for each query to facilitate contrastive learning. For classification and visual grounding tasks, the false negative filtering threshold is set to $\alpha = 0.8$; for VQA and retrieval tasks, we set $\alpha = 0.7$.

### 4.2 MAIN RESULT

### 4.2.1 MULTI-MODAL RETRIEVAL

In Table 1, we present the quantitative evaluation results of the proposed ReCo on the MMEB benchmark (Jiang et al., 2024c), where our method achieves state-of-the-art performance. Specifically, with Qwen2-VL as the base model, ReCo attains an overall score of 73.9, surpassing the previous

Table 1: Results on the MMEB benchmark (Jiang et al., 2024c). IND represents the in-distribution dataset, and OOD represents the out-of-distribution dataset. The reported scores are the average Precision@1 over the corresponding datasets. The best results are marked in bold, and the second-best results are underlined.

| Model | Per Meta-Task Score | | | | Average Score | | |
|---|---|---|---|---|---|---|---|
| | Classification | VQA | Retrieval | Grounding | IND | OOD | Overall |
| # of Datasets → | 10 | 10 | 12 | 4 | 20 | 16 | 36 |
| *CLIP-series Models* | | | | | | | |
| CLIP (Radford et al., 2021) | 42.8 | 9.1 | 53.0 | 51.8 | 37.1 | 38.7 | 37.8 |
| BLIP2 (Li et al., 2023) | 27.0 | 4.2 | 33.9 | 47.0 | 25.3 | 25.1 | 25.2 |
| SigLIP (Zhai et al., 2023) | 40.3 | 8.4 | 31.6 | 59.5 | 32.3 | 38.0 | 34.8 |
| OpenCLIP (Cherti et al., 2023) | 47.8 | 10.9 | 52.3 | 53.3 | 39.3 | 40.2 | 39.7 |
| Magiclens (Zhang et al., 2024c) | 38.8 | 8.3 | 35.4 | 26.0 | 31.0 | 23.7 | 27.8 |
| EVA-CLIP (Sun et al., 2023) | 56.0 | 10.4 | 49.2 | 58.9 | 38.1 | 45.6 | 43.7 |
| *MLLM-based Models* | | | | | | | |
| E5-V (LLaVA-1.6-7B) (Jiang et al., 2024b) | 39.7 | 10.8 | 39.4 | 60.2 | 34.2 | 33.4 | 33.9 |
| VLM2Vec (LLaVA-1.6-7B) (Jiang et al., 2024c) | 61.2 | 49.9 | 67.4 | 86.1 | 67.5 | 57.1 | 62.9 |
| VLM2Vec (Qwen2-VL-7B) (Jiang et al., 2024c) | 62.6 | 57.8 | 69.9 | 81.7 | 72.2 | 57.8 | 65.8 |
| MMRet (LLaVA-1.6-7B) (Zhou et al., 2024) | 56.0 | 57.4 | 69.9 | 83.6 | 68.0 | 59.1 | 64.1 |
| UniME (LLaVA-1.6-7B) (Gu et al., 2025) | 60.6 | 52.9 | 67.9 | 85.1 | 68.4 | 57.9 | 66.6 |
| UniME (LLaVA-OV-7B) (Gu et al., 2025) | 66.8 | 66.6 | 70.5 | 90.9 | 74.6 | 65.8 | 70.7 |
| CAFe (LLaVA-OV-7B) (Yu et al., 2025) | 65.2 | 65.6 | 70.0 | 91.2 | 75.8 | 62.4 | 69.8 |
| mmE5 (Llama-3.2-Vision-11B) (Chen et al., 2025) | 67.6 | 62.8 | 70.9 | 89.7 | 72.3 | 66.7 | 69.8 |
| IDMR (InternVL2.5-26B) (Liu et al., 2025a) | 66.3 | 61.9 | 71.1 | 88.6 | 73.4 | 63.9 | 69.2 |
| LLaVE (Llava-OV-7B) (Lan et al., 2025) | 65.7 | 65.4 | 70.9 | **91.9** | 75.0 | 64.4 | 70.3 |
| UNITE (Qwen2-VL-7B) (Kong et al., 2025) | 68.3 | 65.1 | 71.6 | 84.8 | 73.6 | 66.3 | 70.3 |
| B3 (Qwen2-VL-7B) (Thirukovalluru et al., 2025) | 70.0 | 66.5 | **74.1** | 84.6 | 75.9 | 67.1 | 72.0 |
| QQMM-embed (LLaVA-OV-7B) (Xue et al., 2025) | 66.8 | 66.8 | 70.5 | 90.4 | 74.7 | 65.6 | 70.7 |
| QQMM-embed (QQMM-7B) (Xue et al., 2025) | 69.9 | 70.0 | 72.1 | 86.0 | 77.2 | 66.6 | 72.5 |
| Ours (Qwen2-VL-7B) | **71.0** | **71.5** | 73.7 | 87.7 | **77.6** | **69.2** | **73.9** |

SOTA model QQMM-embed (Xue et al., 2025) by 1.4 points and outperforming B3 (Thirukovalluru et al., 2025), which also uses Qwen2-VL as its backbone, by 1.9 points. Moreover, ReCo demonstrates superior performance on the VQA task, outperforming the next best model (QQMM-embed) by 1.5 points, which we attribute to the proposed autoregressive reconstruction task that helps preserve the powerful instruction-following ability of MLLMs. Notably, our method also achieves an impressive score of 69.2 on out-of-distribution datasets, exceeding the next best model (B3) by 2.1 points, indicating strong generalization ability. These significant performance improvements are driven by the proposed autoregressive reconstruction task and the hard negative augmentation training strategy, which together foster a discriminative unified representation space enriched with semantic information, thereby enhancing the discriminability of the learned representations.

### 4.2.2 ZERO-SHOT CROSS-MODAL RETRIEVAL

Here, we analyze the zero-shot cross-modal retrieval capability of our method. First, we conduct experiments on fine-grained cross-modal retrieval datasets Urban1K (Zhang et al., 2024a) and DOCCI (Onoe et al., 2024), featured by detailed textual descriptions. As shown in Table 2, our method outperforms UNITE (Kong et al., 2025)—a strong unified embedding model—on image-to-text retrieval in Urban1K, as well as on both text-to-image and image-to-text retrieval tasks in DOCCI. Our proposed autoregressive reconstruction task requires the model to generate the corresponding instance semantic content from embeddings, thereby compelling the embeddings to encode both individual elements and their interrelations. This encourages the preservation of MLLM's inherent strengths in compositional understanding. Notably, due to the highly compositional and fine-grained descriptions in DOCCI (e.g., precise spatial relations and accurate object counts), embeddings trained solely with contrastive learning exhibit limitations in compositional understanding, as it does not explicitly enforce modeling the relationships among individual elements. Consequently, our method achieves significant performance improvements on DOCCI, surpassing UNITE by 5.3 points on text-to-image retrieval and by 7.1 points on image-to-text retrieval. For coarse-grained

Table 2: Results of fine-grained image-text retrieval on Urban1K (Zhang et al., 2024a) and DOCCI (Onoe et al., 2024).

Table 3: Zero-shot coarse-grained image-text retrieval results on Flickr30K (Plummer et al., 2015).

| Model | Urban1K | | DOCCI | |
|---|---|---|---|---|
| | T→I | I→T | T→I | I→T |
| *CLIP-based Models* | | | | |
| Long-CLIP (Zhang et al., 2024a) | 86.1 | 82.7 | 78.6 | 66.5 |
| FineLIP (Asokan et al., 2025) | 94.1 | 93.2 | 86.0 | 84.5 |
| *MLLM-based Models* | | | | |
| MATE (Jang et al., 2024) | - | - | 84.6 | 76.6 |
| E5-V-7B (Jiang et al., 2024b) | 88.9 | 83.2 | - | - |
| VLM2Vec-7B (Jiang et al., 2024c) | 90.8 | 84.7 | - | - |
| UniME-7B (Gu et al., 2025) | 95.2 | 95.9 | - | - |
| UNITE-7B (Kong et al., 2025) | **95.5** | 95.6 | 87.2 | 85.8 |
| Ours-7B | 95.3 | **98.4** | **92.5** | **92.9** |

| Model | Flickr30K | |
|---|---|---|
| | T→I | I→T |
| *CLIP-based Models* | | |
| OpenCLIP (Cherti et al., 2023) | 75.0 | 88.7 |
| MagicLens (Zhang et al., 2024c) | 79.7 | 89.6 |
| *MLLM-based Models* | | |
| CAFe-7B (Yu et al., 2025) | 75.3 | 87.5 |
| VLM2Vec-7B (Jiang et al., 2024c) | 80.3 | 94.6 |
| LamRA-7B (Liu et al., 2025b) | 82.8 | 92.7 |
| UniME-7B (Gu et al., 2025) | 81.9 | 93.4 |
| UNITE-7B (Kong et al., 2025) | **86.1** | 94.4 |
| Ours-7B | 85.5 | **95.9** |

Table 4: Ablation results on MMEB. The model with ID 0 is optimized using the InfoNCE loss.

| ID | Model | Batch size | IND | OOD | Overall |
|---|---|---|---|---|---|
| 0 | Baseline (Qwen2-VL-7B + InfoNCE) | 1024 | 74.2 | 66.0 | 70.5 |
| 1 | 0 + Hard negative mining | 256 | 75.9 | 67.2 | 72.0 |
| 2 | 1 + Unfreeze projector | 256 | 76.5 | 67.9 | 72.7 |
| 3 | 2 + Unfreeze image encoder | 256 | 76.2 | 66.8 | 72.0 |
| 4 | 1 + Unfreeze projector | 512 | 77.2 | 68.4 | 73.3 |
| 5 | 4 + False negative filtering | 512 | 77.1 | 69.0 | 73.5 |
| 6 | 5 + Autoregressive task (without the modified attention mask) | 512 | 77.3 | 68.8 | 73.5 |
| 7 | 5 + Autoregressive reconstruction task | 512 | **77.6** | **69.2** | **73.9** |

cross-modal retrieval on the Flickr30K (Plummer et al., 2015), which features short textual captions, as shown in Table 3, our model achieves strong retrieval performance on both text-to-image and image-to-text tasks, demonstrating its effectiveness in aligning text and image modalities.

### 4.3 ABLATION STUDY

Table 4 presents an ablation study analyzing the impact of different design choices on the performance of ReCo across IND, OOD, and overall metrics. As the reference setting, the baseline model (ID 0) fine-tunes the LLM backbone via LoRA using the standard InfoNCE with batch size of 1024.

**Hard negative mining is crucial for effective contrastive learning.** To ensure a fair comparison, we reduce the batch size of model (ID 1) to 256 and mine three hard negatives per query, thereby keeping the number of candidates consistent with the baseline. As shown in Table 4 ($0 \rightarrow 1$), incorporating the hard negative mining strategy yields significant improvements on both IND (+1.7) and OOD (+1.2), resulting in a notable overall gain (+1.5). This improvement arises because hard negatives provide richer gradient signals during contrastive learning.

**Unfreezing the projector facilitates adapting the MLLM to an embedding model.** Unfreezing the projector ($1 \rightarrow 2$) yields an overall gain (+0.7), as adapting an MLLM to an embedding model requires re-adaptation of the representation space. In contrast, unfreezing the image encoder ($2 \rightarrow 3$) degrades OOD performance (–1.1), since instruction tuning on a smaller dataset may weaken generalization. To maintain a comparable training budget, we increase the batch size to 512 ($2 \rightarrow 4$), keeping the total in-batch samples consistent with the baseline, yielding another gain (+0.6).

**False negative filtering improves the model's generalization ability.** The presence of false negatives within a batch hinders the effective discrimination of hard negatives during contrastive learning. As shown in Table 4 ($4 \rightarrow 5$), applying the false negative filtering mechanism improves OOD performance (+0.6). This improvement arises because filtering out false negatives alleviates contamination from mislabeled positives, encouraging the model to learn higher-level features with rich semantic content rather than overfitting to idiosyncrasies of the training set, thereby enhancing generalization.

**The autoregressive reconstruction task further enhances representation discriminability.** Our proposed autoregressive reconstruction task improves the quality of learned embeddings ($5 \rightarrow 7$), yielding performance gains on IND (+0.5) and OOD (+0.2). Notably, applying the autoregressive objective without the modified attention mask in Figure 2—i.e., without performing the actual reconstruction task—fails to deliver any improvement ($5 \rightarrow 6$). This is because a standard autoregressive task does not explicitly guide embeddings to capture rich semantic information from the input, whereas the reconstruction task does, thereby producing embeddings enriched with semantic information. These results empirically validate the effectiveness of our proposed autoregressive reconstruction task.

### 4.4 PRINCIPAL COMPONENT ANALYSIS

In Figure 3, we analyze the impact of the autoregressive reconstruction task on semantic richness and

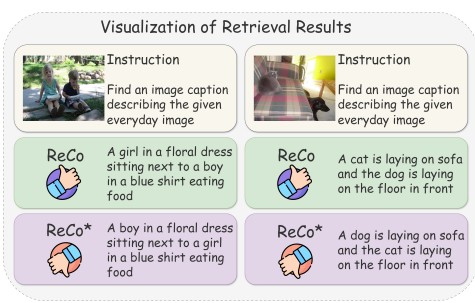

Figure 3: Relationship between semantic richness and discriminability of embeddings at different training steps (500, 1000, 1500, 2000) on the MSCOCO_t2i evaluation set, where ReCo* denotes models optimized with contrastive learning alone.

discriminability of embeddings. Specifically, we report the relationship between model performance and the number of principal components required to explain 80% of the total variance at different training steps (500, 1000, 1500, 2000) on the MSCOCO_t2i dataset. The results show that, compared to models trained solely with contrastive learning, the autoregressive reconstruction task enables embeddings to retain more semantic information about the input, resulting in a more evenly distributed variance across its principal components, thereby exhibiting higher semantic richness and stronger discriminability.

### 4.5 VISUALIZATION OF RETRIEVAL RESULTS

In Figure 4, we present qualitative comparison of retrieval results between ReCo and ReCo* (optimized with contrastive learning alone). On the left, ReCo successfully retrieves the correct description, "A girl in a floral dress sitting next to a boy in a blue shirt eating food," whereas ReCo* incorrectly swaps the genders. In the right example, ReCo retrieves the accurate description, "A cat is laying on sofa and the dog is laying on the floor in front," while ReCo* mistakenly reverses the spatial relationship, describing the dog as being on the sofa and the cat on the floor. Additional visualization examples are provided in Figure 5. These results demonstrate that our proposed autoregressive reconstruction task promotes embeddings to retain richer semantic details,

Figure 4: Visualization of retrieval results, where ReCo* denotes models optimized with contrastive learning alone.

such as entity attributes and spatial relations, thereby producing more discriminative embeddings enriched with semantic information and consequently improving performance on downstream tasks.

## 5 CONCLUSION

In this work, we propose ReCo, a novel unified embedding framework for learning high-quality representations that are simultaneously discriminative and information-rich. To ensure that embeddings retain rich semantic information from the input, we propose an autoregressive reconstruction task, which forces the model to reconstruct the instance semantic content from the corresponding embeddings, thereby shaping a unified representation space. Furthermore, we introduce a hard negative augmentation training strategy to enhance the model's ability to distinguish positives from hard negatives by mining challenging negatives and mitigating false negative contamination. Extensive experiments demonstrate the superiority of ReCo and the effectiveness of its key components.

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

# A DATA

## A.1 MMEB BENCHMARK

MMEB (Jiang et al., 2024c) is a comprehensive multimodal embedding benchmark that spans diverse domains (e.g., common, news, Wikipedia, web, and fashion) and supports various instruction types, from object recognition to retrieval tasks. As shown in Table 5, MMEB consists of 36 datasets across four meta-tasks: classification, visual question answering, retrieval, and visual grounding. The benchmark is strategically divided into 20 in-distribution training datasets and 16 out-of-distribution evaluation datasets. All tasks are formulated as ranking problems, where the model is required to process instruction-guided queries (text, image, or both) to identify the correct target from a set of candidates. Performance is evaluated using Precision@1, which measures the percentage of top candidate matching the groundtruth. This comprehensive design makes MMEB an ideal testbed for developing and assessing universal multimodal embeddings. Specifically, MMEB is organized into four primary meta-task categories, which are structured as follows:

Table 5: The statistics of MMEB (Jiang et al., 2024c): 36 datasets across 4 meta-task categories, with 20 in-distribution datasets used for training and 16 out-of-distribution datasets used exclusively for evaluation. A checkmark in the Reconstruction column denotes joint optimization with autoregressive reconstruction and contrastive learning.

| Meta-Task | Dataset | Query→Target | Distribution Type | Reconstruction | #Training | #Eval | #Candidates |
|---|---|---|---|---|---|---|---|
| Classification (10 Tasks) | ImageNet-1K | I→T | IND | ✓ | 100K | 1000 | 1000 |
| | N24News | I + T→T | IND | ✓ | 49K | 1000 | 24 |
| | HatefulMemes | I→T | IND | ✓ | 8K | 1000 | 2 |
| | VOC2007 | I→T | IND | ✓ | 8K | 1000 | 20 |
| | SUN397 | I→T | IND | ✓ | 20K | 1000 | 397 |
| | Place365 | I→T | OOD | | - | 1000 | 365 |
| | ImageNet-A | I→T | OOD | | - | 1000 | 1000 |
| | ImageNet-R | I→T | OOD | | - | 1000 | 200 |
| | ObjectNet | I→T | OOD | | - | 1000 | 313 |
| | Country-211 | I→T | OOD | | - | 1000 | 211 |
| VQA (10 Tasks) | OK-VQA | I + T→T | IND | ✓ | 9K | 1000 | 1000 |
| | A-OKVQA | I + T→T | IND | ✓ | 17K | 1000 | 1000 |
| | DocVQA | I + T→T | IND | ✓ | 40K | 1000 | 1000 |
| | InfographicVQA | I + T→T | IND | ✓ | 24K | 1000 | 1000 |
| | ChartQA | I + T→T | IND | ✓ | 28K | 1000 | 1000 |
| | Visual7W | I + T→T | IND | ✓ | 70K | 1000 | 1000 |
| | ScienceQA | I + T→T | OOD | | - | 1000 | 1000 |
| | VizWiz | I + T→T | OOD | | - | 1000 | 1000 |
| | GQA | I + T→T | OOD | | - | 1000 | 1000 |
| | TextVQA | I + T→T | OOD | | - | 1000 | 1000 |
| Retrieval (12 Tasks) | VisDial | T→I | IND | | 123K | 1000 | 1000 |
| | CIRR | I + T→I | IND | | 26K | 1000 | 1000 |
| | VisualNews_t2i | T→I | IND | ✓ | 100K | 1000 | 1000 |
| | VisualNews_i2t | I→T | IND | ✓ | 100K | 1000 | 1000 |
| | MSCOCO_t2i | T→I | IND | ✓ | 100K | 1000 | 1000 |
| | MSCOCO_i2t | I→T | IND | ✓ | 113K | 1000 | 1000 |
| | NIGHTS | I→I | IND | | 16K | 1000 | 1000 |
| | WebQA | T→I + T | IND | | 17K | 1000 | 1000 |
| | OVEN | I + T→I + T | OOD | | - | 1000 | 1000 |
| | FashionIQ | I + T→I | OOD | | - | 1000 | 1000 |
| | EDIS | T→I + T | OOD | | - | 1000 | 1000 |
| | Wiki-SS-NQ | T→I | OOD | | - | 1000 | 1000 |
| Visual Grounding (4 Tasks) | MSCOCO | I + T→I | IND | | 100K | 1000 | 1000 |
| | Visual7W-Pointing | I + T→I | OOD | | - | 1000 | 1000 |
| | RefCOCO | I + T→I | OOD | | - | 1000 | 1000 |
| | RefCOCO-Matching | I + T→I + T | OOD | | - | 1000 | 1000 |

Table 6: The statistics of Urban1K (Zhang et al., 2024a), DOCCI (Onoe et al., 2024), and Flickr30K (Plummer et al., 2015). # Queries represents the number of test queries, and # Candidates denotes the number of test candidates per query.

| Benchmark | Query→Target | Zero-shot | #Queries | #Candidates |
|---|---|---|---|---|
| Urban1K (Zhang et al., 2024a) | T→I, I→T | ✓ | 1,000 | 1,000 |
| DOCCI (Onoe et al., 2024) | T→I, I→T | ✓ | 5,000 | 5,000 |
| Flickr30K (Plummer et al., 2015) | T→I, I→T | ✓ | 1,000 | 5,000 |

Table 7: Additional task instructions for candidates that are used in ReCo. For candidates from other datasets, we use the existing instructions from VLM2Vec (Jiang et al., 2024c). These instructions are expected to facilitate the disentanglement of different tasks during training.

| Category | Dataset | Task Instruction |
|---|---|---|
| Retrieval | MSCOCO_i2t VisualNews_i2t | Represent the following answer to an image caption task: |
| Classification | ImageNet-1K HatefulMemes SUN397 N24News VOC2007 Place365 ImageNet-A ImageNet-R ObjectNet Country211 | Represent the following answer to an image classification task: |
| Visual Question Answering | OK-VQA A-OKVQA DocVQA InfographicsVQA ChartQA Visual7W ScienceQA GQA TextVQA VizWiz | Represent the following answer to a VQA task: |

1. Classification: The query consists of an instruction, an image, optionally accompanied by related text, while the target is the class name. The number of candidates equals the number of classes.

2. Visual question answering: The query consists of an instruction, an image, and a piece of text as the question, while the target is the answer.

3. Retrieval: Both the query and target may involve combinations of text, image, and instruction.

4. Visual grounding: The task is adapted from object detection. The query consists of an instruction with the full image to locate a specific object, and the target is the cropped region of the object.

## A.2 CROSS-MODAL RETRIEVAL DATASETS

In Table 6, we provide a brief description of the cross-modal retrieval datasets, including fine-grained cross-modal retrieval (Urban1K (Zhang et al., 2024a) and DOCCI (Onoe et al., 2024)) and coarse-grained cross-modal retrieval (Flickr30K (Plummer et al., 2015)). We evaluate on each dataset using its standard protocol, with Recall@1 as the primary metric.

Urban1K (Zhang et al., 2024a) is derived from Urban-200 and expanded to 1,000 image–text pairs of urban scenes. Each image is paired with a GPT-4V–generated caption (averaging 101 words) that provides detailed descriptions of object types, colors, and spatial relations. The dataset challenges models with visually similar urban scenes, requiring fine-grained cross-modal understanding.

DOCCI (Onoe et al., 2024) (Descriptions of Connected and Contrasting Images) contains 15,000 images paired with human-annotated descriptions averaging 136 words. Curated by a single researcher, the dataset evaluates spatial relations, counting, text rendering, and world knowledge comprehension. It features contrast sets in which object arrangements differ subtly. For evaluation, we follow the official 5K test split.

Flickr30K (Plummer et al., 2015) consists of 31K images collected from Flickr, with each image paired with five human-annotated captions describing its content. The dataset covers a wide range of everyday scenes and human activities in natural environments. Following common practice (Karpathy & Fei-Fei, 2015), we use the standard split with 1,000 images for testing.

### A.3 TASK INSTRUCTION

The MMEB benchmark (Jiang et al., 2024c) provides task-specific instructions for queries and candidates in most datasets. However, for inputs involving text-only candidates, such instructions are not available. Table 7 summarizes the task instructions used for text-only candidate inputs in our experiments. Most of these tasks fall under classification and visual question answering.

For the cross-modal retrieval tasks (Urban1K (Zhang et al., 2024a), DOCCI (Onoe et al., 2024), and Flickr30K (Plummer et al., 2015)), we design unified task instructions based on the query-to-candidate modality. Specifically, for text-to-image retrieval, the task instructions for the query and candidate sides are `"Find me an everyday image that matches the given caption:"` and `"Represent the given image."`, respectively, and for image-to-text retrieval, the corresponding instructions are `"Find an image caption describing the given everyday image."` and `"Represent the following answer to an image caption task:"`.

## B MORE EXPERIMENTAL RESULTS

### B.1 DETAILED MAIN RESULTS

Table 8 provides the detailed results of ReCo and baseline models with full scores reported across the 36 tasks in MMEB.

### B.2 FURTHER VISUALIZATIONS OF RETRIEVAL RESULTS

Figure 5 shows additional qualitative comparisons of retrieval results between ReCo and ReCo* (optimized with contrastive learning alone).

## C LIMITATION AND FUTURE WORK

ReCo is trained only on embedding datasets involving arbitrary combinations of text and image modalities, where it demonstrates superior performance across both. However, with the rapid growth of social media, embedding models for video and audio modalities have become critical for tasks such as search and recommendation. In future work, we plan to extend ReCo to more general multimodal scenarios and further explore a unified framework integrating embedding and generation.

## D DISCUSSION OF ETHICS

To the best of our knowledge, the datasets used in this work do not contain any sensitive information. However, in future real-world applications, our model may be deployed in scenarios involving potentially sensitive domains, such as healthcare, education, or public safety. Therefore, we emphasize that before applying this approach in specific environments, practitioners should conduct thorough compliance reviews and carry out systematic safety validation and robustness testing to ensure its reliability and security in practice.

## E DECLARATION OF LLM USAGE

LLM is used only for writing, editing, or formatting purposes.

Table 8: The detailed results of the baselines and ReCo on MMEB, which includes 20 in-distribution (IND) datasets and 16 out-of-distribution (OOD) datasets. The out-of-distribution datasets are highlighted with a yellow background in the table. For each model, we report its best variant with fully available performance metrics, including CLIP (Radford et al., 2021), VLM2Vec (LLaVA-1.6-7B) (Jiang et al., 2024c), MMRet (LLaVA-1.6-7B) (Zhou et al., 2024), UniME (LLaVA-1.6-7B) (Gu et al., 2025), mmE5 (Llama-3.2-Vision-11B) (Chen et al., 2025), IDMR (InternVL2.5-26B) (Liu et al., 2025a), LLaVE (Llava-OV-7B) (Lan et al., 2025), and UNITE (Qwen2-VL-7B) (Kong et al., 2025). Superior versions might exist but are excluded due to incomplete score reporting.

| | CLIP | VLM2Vec | MMRet | UniME | mmE5 | IDMR | LLaVE | UNITE | ReCo |
|---|---|---|---|---|---|---|---|---|---|
| **Classification (10 tasks)** | | | | | | | | | |
| ImageNet-1K | 55.8 | 74.5 | 58.8 | 71.3 | 77.8 | 80.6 | 77.1 | 80.2 | 84.2 |
| N24News | 34.7 | 80.3 | 71.3 | 79.5 | 81.7 | 81.6 | 82.1 | 80.3 | 83.8 |
| HatefulMemes | 51.1 | 67.9 | 53.7 | 64.6 | 64.2 | 72.3 | 74.3 | 67.1 | 73.6 |
| VOC2007 | 50.7 | 91.5 | 85.0 | 90.4 | 91.0 | 92.7 | 90.3 | 84.9 | 88.8 |
| SUN397 | 43.4 | 75.8 | 70.0 | 75.9 | 77.7 | 78.8 | 79.1 | 78.7 | 81.2 |
| Place365 | 28.5 | 44.0 | 43.0 | 45.6 | 43 | 38.9 | 45.1 | 44.5 | 47.4 |
| ImageNet-A | 25.5 | 43.6 | 36.1 | 45.5 | 56.3 | 63.6 | 51.6 | 59.2 | 58.3 |
| ImageNet-R | 75.6 | 79.8 | 71.6 | 78.4 | 86.3 | 84 | 90.9 | 90.5 | 90.1 |
| ObjectNet | 43.4 | 39.6 | 55.8 | 36.4 | 62.5 | 50.5 | 46.2 | 68.1 | 74.1 |
| Country-211 | 19.2 | 14.7 | 14.7 | 18.7 | 35.4 | 20.3 | 20.1 | 29.5 | 28.0 |
| *All Classification* | 42.8 | 61.2 | 56.0 | 60.6 | 67.6 | 66.3 | 65.7 | 68.3 | 71.0 |
| **VQA (10 tasks)** | | | | | | | | | |
| OK-VQA | 7.5 | 69.0 | 73.3 | 68.3 | 67.6 | 71.0 | 71.1 | 67.1 | 74.1 |
| A-OKVQA | 3.8 | 54.4 | 56.7 | 58.7 | 56.1 | 59.2 | 70.8 | 58.0 | 61.8 |
| DocVQA | 4.0 | 52.0 | 78.5 | 67.6 | 90.3 | 75.1 | 90.3 | 92.7 | 95.1 |
| InfographicsVQA | 4.6 | 30.7 | 39.3 | 37.0 | 56.5 | 44.6 | 53.5 | 71.3 | 76.3 |
| ChartQA | 1.4 | 34.8 | 41.7 | 33.4 | 50.5 | 64.6 | 62.2 | 63.2 | 66.7 |
| Visual7W | 4.0 | 49.8 | 49.5 | 51.7 | 51.9 | 54.9 | 55.8 | 54.9 | 67.2 |
| ScienceQA | 9.4 | 42.1 | 45.2 | 40.5 | 55.8 | 54.7 | 54.4 | 51.2 | 54.5 |
| VizWiz | 8.2 | 43.0 | 51.7 | 42.7 | 52.8 | 47.1 | 48.5 | 53.4 | 55.4 |
| GQA | 41.3 | 61.2 | 59.0 | 63.6 | 61.7 | 71.0 | 68.4 | 56.8 | 76.8 |
| TextVQA | 7.0 | 62.0 | 79.0 | 65.2 | 83.3 | 77.0 | 79.4 | 82.3 | 87.3 |
| *All VQA* | 9.1 | 49.9 | 57.4 | 52.9 | 62.6 | 61.9 | 65.4 | 65.1 | 71.5 |
| **Retrieval (12 tasks)** | | | | | | | | | |
| VisDial | 30.7 | 80.9 | 83.0 | 79.7 | 74.1 | 81.5 | 83.0 | 80.5 | 85.3 |
| CIRR | 12.6 | 49.9 | 61.4 | 52.2 | 54.7 | 57.6 | 54.5 | 51.6 | 60.7 |
| VisualNews_t2i | 78.9 | 75.4 | 74.2 | 74.8 | 77.6 | 78.5 | 76.6 | 79.3 | 81.5 |
| VisualNews_i2t | 79.6 | 80.0 | 78.1 | 78.8 | 83.3 | 80.6 | 81.2 | 82.4 | 84.3 |
| MSCOCO_t2i | 59.5 | 75.7 | 78.6 | 74.9 | 76.4 | 79.1 | 78.9 | 78.2 | 79.5 |
| MSCOCO_i2t | 57.7 | 73.1 | 72.4 | 73.8 | 73.2 | 75.4 | 74.7 | 74.3 | 74.0 |
| NIGHTS | 60.4 | 65.5 | 68.3 | 66.2 | 68.3 | 68.6 | 67.0 | 66.0 | 68.7 |
| WebQA | 67.5 | 87.6 | 90.2 | 89.8 | 88.0 | 89.0 | 90.4 | 87.0 | 90.7 |
| FashionIQ | 11.4 | 16.2 | 54.9 | 16.5 | 28.8 | 21.0 | 23.3 | 26.3 | 20.6 |
| Wiki-SS-NQ | 55.0 | 60.2 | 24.9 | 66.6 | 65.8 | 66.9 | 63.9 | 72.2 | 72.1 |
| OVEN | 41.1 | 56.5 | 87.5 | 55.7 | 77.5 | 67.4 | 68.0 | 73.1 | 74.0 |
| EDIS | 81.0 | 87.8 | 65.6 | 86.2 | 83.7 | 87.6 | 89.1 | 88.3 | 92.6 |
| *All Retrieval* | 53.0 | 67.4 | 69.9 | 67.9 | 71.0 | 71.1 | 70.9 | 71.6 | 73.7 |
| **Visual Grounding (4 tasks)** | | | | | | | | | |
| MSCOCO | 33.8 | 80.6 | 76.8 | 76.5 | 53.7 | 81.5 | 87.0 | 73.9 | 74.1 |
| RefCOCO | 56.9 | 88.7 | 89.8 | 89.3 | 92.7 | 91.7 | 95.4 | 89.2 | 93.5 |
| RefCOCO-matching | 61.3 | 84.0 | 90.6 | 90.6 | 88.8 | 88.1 | 92.8 | 90.1 | 94.1 |
| Visual7W-pointing | 55.1 | 90.9 | 77.0 | 84.1 | 92.3 | 93.1 | 92.5 | 86.1 | 89.1 |
| *All Visual Grounding* | 51.8 | 86.1 | 83.6 | 85.1 | 89.6 | 88.6 | 91.9 | 84.8 | 87.7 |
| **Final Score (36 tasks)** | | | | | | | | | |
| All | 37.8 | 62.9 | 64.1 | 66.6 | 69.8 | 69.2 | 70.3 | 70.3 | 73.9 |
| All IND | 37.1 | 67.5 | 68.0 | 68.4 | 72.3 | 73.4 | 75.0 | 73.6 | 77.6 |
| All OOD | 38.7 | 57.1 | 59.1 | 57.9 | 66.7 | 63.4 | 64.4 | 66.3 | 69.2 |

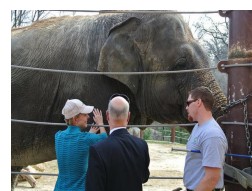

Instruction: Find an image caption describing the given everyday image

ReCo: A woman pats an elephant as a couple men watch

ReCo*: A couple men pat an elephant as a woman watches

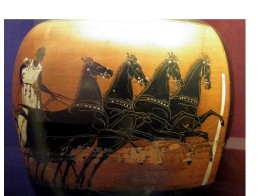

Instruction: Find an image caption describing the given everyday image

ReCo: A brown vase has four black horses on it

ReCo*: A black vase has four brown horses on it

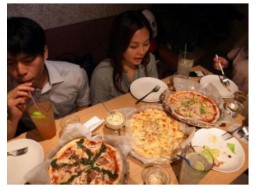

Instruction: Find an image caption describing the given everyday image

ReCo: A group of people sitting at a table holding different pizzas

ReCo*: A young man sitting at a table with a pizza

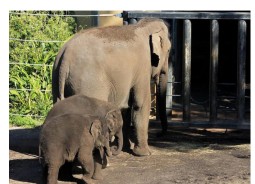

Instruction: Find an image caption describing the given everyday image

ReCo: An elephant leads a baby elephant towards a door

ReCo*: A baby elephant leads an elephant towards a door

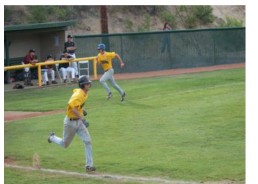

Instruction: Find an image caption describing the given everyday image

ReCo: A couple of men running across a lush green field

ReCo*: A baseball team playing a baseball game on a baseball field

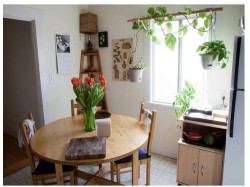

Instruction: Find an image caption describing the given everyday image

ReCo: A bright kitchen with tulips on the table and plants by the window

ReCo*: A bright kitchen with plants on the table and tulips by the window

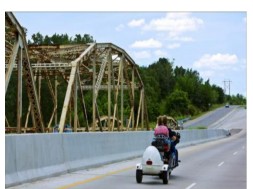

Instruction: Find an image caption describing the given everyday image

ReCo: A couple riding a motorcycle down a street

ReCo*: A bus getting ready to get off a bridge on the highway

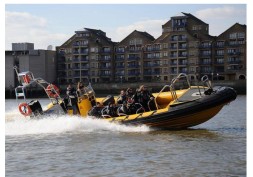

Instruction: Represent the given image for classification

ReCo: speedboat

ReCo*: lifeboat

Figure 5: Further visualizations of retrieval results, where ReCo* denotes models optimized with contrastive learning alone.

