# OpenReview forum: "Universal embeddings via Reconstruction-Augmented Contrastive Learning"
_ICLR.cc/2026/Conference — Submitted to ICLR 2026_

### Official Review · Reviewer_gzez · 2025-10-30

**Soundness:** 2
**Presentation:** 2
**Contribution:** 2
**Rating:** 2
**Confidence:** 3

**Summary:**

This paper proposes ReCo, a reconstruction-augmented contrastive learning framework that enhances multimodal embeddings by enforcing semantic preservation. By jointly training an autoregressive reconstruction task with contrastive learning, ReCo produces embeddings that are both discriminative and semantically rich. Experiments on the MMEB benchmark and cross-modal retrieval tasks show that ReCo achieves state-of-the-art performance.

**Strengths:**

1. Developing universal multimodal embedding models is an important and meaningful task.
2. The proposed approach effectively improves performance across various tasks.

**Weaknesses:**

1. The novelty is limited. Hard negative mining and false negative filtering have been widely used in VLM embedding tasks, such as B3 [1] and QQMM-Embed [2]. Therefore, I believe the main contribution of this work lies in the reconstruction module. However, in the ablation study (Table 4), the performance improvement in experiments 6 and 7 is marginal. Thus, this work appears somewhat incremental.
2. The experiments are only evaluated on Qwen-2.5-VL. The paper would be stronger if the proposed approach were adapted to other baselines, such as LLaVA.
3. I am curious about what the <emb> token specifically represents if it is decoded. Is it a word?


**References**:

[1] Raghuveer Thirukovalluru et al., “Breaking the Batch Barrier (B3) of Contrastive Learning via Smart Batch Mining”, arXiv, 2025.

[2] Xue Youze et al., “Improve Multi-Modal Embedding Learning via Explicit Hard Negative Gradient Amplifying”, arXiv, 2025.

**Questions:**

Please refer to the weaknesses.

---

### Official Review · Reviewer_qK9Y · 2025-10-31

**Soundness:** 2
**Presentation:** 3
**Contribution:** 2
**Rating:** 4
**Confidence:** 4

**Summary:**

Standard contrastive learning does not explicitly impose semantic constraints on the embeddings, which leads to a loss of semantic information and suboptimal retrieval performance. To address this limitation, this paper introduces a novel auxiliary task beyond contrastive learning — an autoregressive reconstruction task — which enforces the model to reconstruct the instance’s semantic content from its embedding. In addition, a hard negative augmentation strategy is proposed to further enhance the classical contrastive learning paradigm. Extensive experiments demonstrate the effectiveness of the proposed method.

**Strengths:**

1. The proposed autoregressive reconstruction task effectively enriches the semantic information of the learned embeddings, serving as a complementary objective to contrastive learning.
2. The proposed hard negative augmentation strategy significantly enhances the effectiveness of contrastive learning.
3. The proposed method achieves state-of-the-art performance on the MMEB benchmark and multiple cross-modal retrieval tasks.

**Weaknesses:**

1.	The abstract does not align well with the main content of the paper.
One of the key contributions of this work is the hard negative augmentation training strategy, which, however, is not reflected or mentioned in the abstract.
2.	 The relationship between Sections 3.2 and 3.3 is unclear.
These two parts appear to be independent and loosely connected. In particular, Section 3.3 seems disjoint from the paper’s main emphasis on semantic information. Please clarify how Section 3.3 contributes to or complements the semantic modeling aspect of your method.
3.	The proposed method involves a number of hyperparameters, such as the hard negative queue length K, the number of random hard negatives k, and the false negative filtering threshold α\alphaα.
However, the experimental section does not include an analysis of their effects. An ablation or sensitivity study on these hyperparameters would greatly strengthen the empirical evaluation.
4.	Minor Weakness：According to Table 1, the proposed method shows strong results on the MMEB benchmark, but exhibits a clear performance gap compared to state-of-the-art methods on the Grounding Meta Task. Could the authors provide some explanation or discussion for this discrepancy?

**Questions:**

Please refer to the Weakness part.

---

### Official Review · Reviewer_Mvzu · 2025-11-02

**Soundness:** 3
**Presentation:** 3
**Contribution:** 2
**Rating:** 4
**Confidence:** 4

**Summary:**

This paper presents ReCo, a reconstruction-augmented contrastive learning framework designed to improve the semantic richness of multimodal embeddings. The idea is conceptually appealing: traditional contrastive objectives optimize for pairwise alignment but often lose fine-grained semantic details, so ReCo adds a reconstruction branch that forces embeddings to recover key semantic attributes (such as class labels, captions, or answers) directly from the embedding token. The method is implemented on Qwen2-VL and evaluated extensively on MMEB and downstream retrieval benchmarks. ReCo achieves improvements over strong baselines like B3, QQMM-Embed, and CAFe, showing better generalization in out-of-domain tasks.

**Strengths:**

- The empirical results are good. ReCo consistently outperforms previous methods on MMEB and multiple retrieval benchmarks, achieving a new state-of-the-art performance. The experimental setup is thorough, covering both in-domain and out-of-domain generalization, and all comparisons are made under a consistent backbone (Qwen2-VL) and fair conditions.

- The paper clearly identifies a real limitation in current contrastive learning that embeddings, though discriminative, often lack semantic richness. Introducing a reconstruction objective to inject semantic supervision is conceptually well-grounded and intuitively appealing.

- The paper is easy to follow and technically precise. The architecture, ablations, and benchmarks are clearly explained, and the evaluation protocol is consistent with recent standards in universal embedding research.

**Weaknesses:**

- While the results are strong, ablation studies (Table 4) indicate that the performance gains mainly come from unfreezing the projector and applying contrastive fine-tuning, rather than the reconstruction objective itself. The reconstruction adds only marginal improvement (+0.4 MMEB), which raises doubts about its true contribution to embedding quality.

- ReCo is conceptually very similar to CAFe, which also integrates contrastive and generative objectives. The paper claims novelty through the “modified attention mask” mechanism, but this alone seems minor. Without a direct, side-by-side comparison under identical conditions or clear analysis of how ReCo changes learning dynamics, the novelty remains limited.

- The main claim is that reconstruction enriches embedding semantics, but there is no direct evidence showing this effect. No semantic probing, latent space visualization, or experiment demonstrates that the learned embeddings are more semantically structured.

- Although the method explicitly introduces a reconstruction objective, the paper never shows qualitative examples of generated outputs (e.g., reconstructed captions or answers). It remains unclear whether the model truly learns to generate semantically meaningful text from embeddings, or if the reconstruction head simply overfits to shallow lexical patterns. Including such qualitative evidence would have made the reconstruction claim far more convincing.

**Questions:**

1. Given that most of the gain comes from unfreezing the projector, can the authors provide clearer evidence (quantitative or qualitative), that the reconstruction loss meaningfully changes the learned embedding space?
2. How does ReCo differ in principle from CAFe beyond the modified attention mask? Were both trained under identical backbones and objectives for a fair comparison?
3. Can the authors include qualitative examples of reconstructed outputs to show that the model indeed learns meaningful generation from embeddings?
4. What is the computational overhead of the reconstruction head, and does it scale efficiently with larger corpora or batch sizes?

---

### Official Review · Reviewer_h679 · 2025-11-08

**Soundness:** 3
**Presentation:** 4
**Contribution:** 3
**Rating:** 4
**Confidence:** 4

**Summary:**

The authors propose a novel universal multimodal embedding method that extends the 'standard' contrastive learning approach for adapting MLLMs to learn embeddings by augmenting with a reconstruction loss, thus learning discriminative representations that also capture semantic representation information in the embedding (referred to as ReCo). ReCo is first motivated by showing that reconstruction ability (i.e., number of principal components) correlates with discriminability of the embeddings (Figure 1). Methodologically, ReCo is is essentially InfoNCE (with some details regarding negative sampling) + \lambda * autoregressive reconstruction loss. The resulting embeddings are evaluated within the VLM2Vec datasets, demonstrating consistent performance improvements across {classification, VQA} tasks and competitive performance on retrieval tasks (and weaker performance on grounding tasks) -- leave to good overall performance and notably good performance on OOD settings. Additionally, on multiple zero-shot cross-modal retrieval datasets, ReCo demonstrates strong text-to-image retrieval performance and state-of-the-art performance on image-to-text retrieval. Ablations studies are performed to show the value of hard negatives in contrastive learning, the effect of unfreezing the projection layer when learning MLLM adaptation to embeddings, the effect of false negative filtering, and the isolated effect of reconstruction loss inclusion. Finally, the authors verify the motivating correlation analysis on the ReCo trained embeddings and qualitative results are presented.

**Strengths:**

Strengths of this submission include:
- The proposed method ReCo is well motivated, conceptually appealing, and results in a (relatively) simple & elegant formulation.
- The experimental results are relatively convincing in that the authors use a strong benchmark, compare against many strong baseline systems, and point out some of the most interesting findings (and provide some interpretation for the findings).
- The zero-shot cross-modal retrieval experiments explore a second important setting and (again), the results support the core claims of the paper.
- The ablation studies do address important questions and provide additional understanding.
- The paper is well-written and I am confident I could replicate most of the experiments as there is more details than most paper submissions.

**Weaknesses:**

Weaknesses of this work include:
- Many of the design choices don't state or consider alternatives. For example, lines 198-203 state adopting the one-word limitation strategy and taking the last token (I think) whereas some works state mean token embedding, etc. are better. Not a big deal, but some discussion like this throughout the paper (and any experiments you may have conducted in the appendices) would demonstrate 'due diligence' in designing the system (even if it doesn't likely affect the core contributions). Conversely, statements in section 3.3 is addressed in the ablation study (even if alternatives weren't considered).
- The empirical results are mixed and it isn't clear that they are alway statistically significant. For the results that are negative, some additional discussion would be useful assuming space permits. For example, the T->I zero-shot experiments seem to warrant discussion and possibly any preliminary observations to motivate future work.
- I haven't looked into the details of MMEB-V2, but at face value, it seems that this could be applied to the video, etc. datasets (a stated limitation of the work). Minimally, if this isn't the case, this should be explicitly mentioned in the limitations.

**Questions:**

Basically, I have three clarification questions:
1. What design alternatives regarding the technical details of training the architecture were considered and are there any experiments conducted to determine the choices (or was it 'just' judgement based on existing literature).
2. Discussion regarding MMEB-V2. This is of particular interest to me with respect to raising my score.
3. Was any sensitivity analysis conducted to determine \lambda=0.2 (lines 313-314)?
4. Any discussion regarding interesting negative results (i.e., where baselines perform better). This is also of interest to me in raising my score.

---

### Meta-Review · Area_Chair_Uwaq · 2026-01-01

**Summary:**

This paper received unanimous negative reviews. Although the reviewers appreciate the reasonable motivation and strong performance of the paper, they raised multiple crucial concerns with incremental novelty (Mvzu, gzez), lack of experimental analysis (e.g., sensitivity analysis on the hyper-parameter (h679, qK9Y), failure case analysis (h679), qualitative analysis (Mvzu), application to other VLMs (gzez)), insufficient discussions on the experimental results (h679, qK9Y), no empirical evidence supporting the main argument (Mvzu), unclear contribution of the proposed method (Mvzu), and presentation issues (qK9Y). The authors did not respond to these comments. The AC found that the concerns are valid and outweigh the strengths of the paper, and thus recommends rejection.

**Reviewer Concerns:**

- Lack of exploration for design alternatives (h679)
- Lack of discussions on the experimental results (h679, qK9Y)
- Lack of discussions regarding MMEB-V2 (h679)
- Lack of sensitivity analysis on the hyper-parameter (h679, qK9Y)
- Lack of failure case analysis (h679)
- Lack of qualitative analysis (Mvzu)
- Unclear contribution of the proposed method (Mvzu)
- Incremental novelty (Mvzu, gzez)
- No empirical evidence supporting the main argument (Mvzu)
- Clarity issues (qK9Y)
	- The abstract does not align well with the major contributions of the paper. (qK9Y)
	- The relationship between Section 3.2 and 3.3 is unclear. (qK9Y)
- Limited experiments (gzez)

**Reviewer Scores:**

The reviewers were unanimously negative and the authors provided no response. Thus it is hard to expect any changes in reviewer scores.

---

### Decision · Program_Chairs · 2026-01-26

Reject